# Environmental conditions regulate the impact of plants on cloud formation

D. F. Zhao[1,*], A. Buchholz[1,*,†], R. Tillmann[1], E. Kleist[2], C. Wu[1], F. Rubach[1,†], A. Kiendler-Scharr[1], Y. Rudich[3], J. Wildt[1,2] & Th. F. Mentel[1]

The terrestrial vegetation emits large amounts of volatile organic compounds (VOC) into the atmosphere, which on oxidation produce secondary organic aerosol (SOA). By acting as cloud condensation nuclei (CCN), SOA influences cloud formation and climate. In a warming climate, changes in environmental factors can cause stresses to plants, inducing changes of the emitted VOC. These can modify particle size and composition. Here we report how induced emissions eventually affect CCN activity of SOA, a key parameter in cloud formation. For boreal forest tree species, insect infestation by aphids causes additional VOC emissions which modifies SOA composition thus hygroscopicity and CCN activity. Moderate heat increases the total amount of constitutive VOC, which has a minor effect on hygroscopicity, but affects CCN activity by increasing the particles' size. The coupling of plant stresses, VOC composition and CCN activity points to an important impact of induced plant emissions on cloud formation and climate.

[1] Institute for Energy and Climate Research, IEK-8: Troposphere, Forschungszentrum Jülich, Jülich 52425, Germany. [2] Institute of Bio- and Geosciences, IBG-2, Forschungszentrum Jülich, Jülich 52425, Germany. [3] Department of Earth and Planetary Sciences, Weizmann Institute of Science, Rehovot 76100, Israel. * These authors contributed equally to this work. † Present addresses: Department of Applied Physics, University of Eastern Finland, 70211 Kuopio, Finland (A.B.); Department of Atmospheric Chemistry, Max-Planck-Institute for Chemistry, Mainz 55128, Germany (F.R.). Correspondence and requests for materials should be addressed to Th.F.M. (email: t.mentel@fz-juelich.de) or to D.F.Z. (email: d.zhao@fz-juelich.de).

Volatile organic compounds (VOC) such as isoprene and terpenes emitted by plants have various biological functions, including plant growth, defense and communication[1,2]. It is estimated that 1,000 Tg of biogenic VOC are emitted globally per year, far exceeding total VOC emissions from human activities[1,3]. The VOC emissions from plants are closely coupled to cloud formation and climate via the formation of secondary organic aerosol (SOA)[4–6] that contribute to the regional and global cloud condensation nuclei (CCN) budget[7] (Fig. 1). VOC emissions are regulated by biotic and abiotic environmental factors. Heat, drought or infestation, are stress factors that cause deviations from the plants' optimal living conditions. Environmental factors and stresses change plant emissions significantly in two ways[8–10]. They can either increase or decrease the amount of constitutive VOC (Fig. 1, left path), or stimulate biochemical pathways that induce the emission of other types of VOC. The latter shifts the overall emission composition[8], that is, the relative contributions of different classes of VOC (Fig. 1, right path). Such changes in emitted VOC result in changes of SOA particle size and chemical composition, hence hygroscopicity. Since both, size and hygroscopicity, determine the CCN activity of SOA[11], changes in VOC will eventually influence cloud formation, and ultimately impact climate[12,13]. Already at present, environmental stress factors strongly affect plants[14,15]. According to regular inspections of forest plots, more than 40% of forest trees in Europe suffer from various stresses where biotic stresses account for ∼40% of the total stresses[14,16]. With climate change, stressors such as heat waves, droughts and infestation are projected to intensify and to more often influence the plants' environmental conditions[15] (Fig. 1). This implies a potential important feedback between plants' emissions and climate[13].

The effects of environmental factors on VOC emissions of plants have been investigated in a number of studies, but significantly less than their effects on the net $CO_2$ exchange of plants[8,9,17]. Only few studies addressed effects of environmental factors on induced VOC emissions and SOA formation[10,12,18]. Importantly, the eventual effects of environmental factors on the CCN activity of SOA and on CCN concentrations are unknown so far, restricting our understanding of how terrestrial plants interact with climate.

We address this gap with a new laboratory study on how plant emissions induced by biotic and abiotic environmental factors modify the hygroscopicity and CCN activity of SOA. We investigated the effect of aphid infestation as an example of biotic stresses and the effect of heat and drought as examples of abiotic stresses for both constitutive emissions and induced emissions. VOC emissions were taken from a mixed stand of pine, spruce, and birch, typical for boreal forests (thereafter referred to as boreal trees) and of individual trees of the same types kept in the Jülich Plant Atmosphere Chamber (JPAC)[19] under varying conditions. SOA was formed by photochemical oxidation of constitutive and induced VOC (via homogeneous nucleation) and the CCN activity of the SOA particles was directly determined (see Methods section and Supplementary Fig. 1). It is found that insect infestation, as an example of biotic factors, caused additional VOC emissions which modified SOA composition thus hygroscopicity and CCN activity. And heat, as an example of abiotic factors, increased the total amount of constitutive VOC emissions, which had a minor effect on the hygroscopicity of SOA, but affected CCN activity by increasing the particles' size.

## Results

**Effect of insect infestation.** The constitutive VOC emissions of boreal trees in the absence of stresses are dominated by

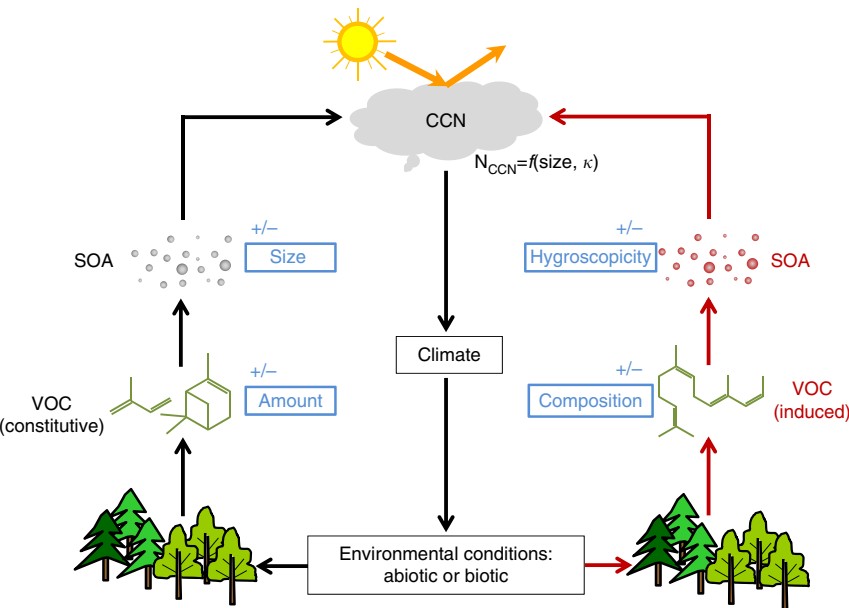

**Figure 1 | Interactions of plant emissions and cloud formation.** The schematic shows the interactions of environmental conditions, plant volatile organic compound (VOC) emissions, secondary organic aerosol (SOA), cloud formation and climate. In unstressed conditions, plants emit constitutive VOC (black arrows on the left path), which on oxidation form SOA that act as cloud condensation nuclei (CCN) and can affect cloud formation and climate. Unfavourable environmental conditions (stresses) can induce VOC emissions (red arrows on the right path). Climatic changes and the resulting environmental conditions can affect the amount of constitutive VOC emissions and/or induce VOC emissions that modify the VOC composition. Such alterations in VOC emissions will be reflected in the particle size and/or particle composition. The latter determines the hygroscopicity parameter ($\kappa$) of the SOA, which is a measure of CCN activity at a given particle size. Both, particle size and $\kappa$, determine the CCN number concentration ($N_{CCN}$) (cf. Supplementary Fig. 5), and thus affect cloud formation and climate. $+/-$ indicates the changes of parameters.

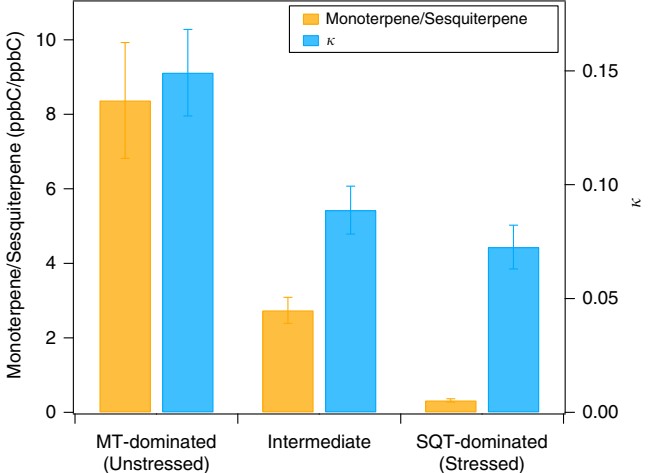

**Figure 2 | Volatile organic compounds composition and hygroscopicity parameter κ.** The ratio of monoterpene (MT) to sesquiterpene (SQT) emissions (ppbC/ppbC, yellow bar, left axis) from unstressed trees (monoterpene-dominated emissions) and biotically stressed (aphid infestation) boreal trees (sesquiterpene-dominated emissions) and the corresponding κ value (blue bar, right axis) of the resulting secondary organic aerosol (SOA) are shown. The emissions from unstressed trees are from pines here. The 'intermediate' case was obtained for the emissions of stressed boreal trees in the dark, when fractions of monoterpenes and sesquiterpenes were between those of monoterpene-dominated and sesquiterpene-dominated cases. The emissions here were obtained at room temperature (22–25 °C). The error bars represent the standard deviations of measurements for κ and volatile organic compounds concentrations (see Supplementary Table 3 for detailed data).

monoterpenes[19] (see also Supplementary Fig. 2). Under biotic stress such as insect infestation, the composition of emitted VOC changes significantly and is often dominated by induced VOC such as sesquiterpenes and green leaf volatiles[2,8,18]. The CCN activity of SOA is characterized using the hygroscopicity parameter κ, wherein higher κ indicates higher CCN activity for particles of a given size. For comparison, the highly water-soluble inorganic salt ammonium sulfate has a κ of ∼0.6 and highly water insoluble and non-wettable black carbon has a κ of 0. κ was ∼0.15 for SOA from monoterpene-dominated VOC emissions from unstressed pines, composed of 78% monoterpenes and 8% sesquiterpenes (Fig. 2, 'MT-dominated' case). During the measurements with insect infested boreal trees, sesquiterpenes dominated the VOC emissions (72% of the total carbon). Concomitant with the shift of the emission composition from monoterpene-dominated to sesquiterpene-dominated, κ of SOA decreased significantly from 0.15 ± 0.02 to 0.07 ± 0.01 (Fig. 2). Particles from VOC emissions with moderate sesquiterpene fraction (monoterpenes 70% versus sesquiterpenes 27%, Fig. 2, 'intermediate' case) have κ values in the middle of the monoterpene- and sesquiterpene-dominated cases. While the monoterpene emission was more than double of the sesquiterpene emission for the intermediate case, the amount of sesquiterpene oxidation products in the particles was estimated to be similar to that of monoterpene oxidation products, since the particle yield of sesquiterpene oxidation is substantially higher than that of monoterpene oxidation (17% versus 5%)[18,19] (see Methods section). Therefore, the CCN activity was still significantly reduced compared with the unstressed monoterpene-dominated case. As shown below, such changes of κ can significantly affect the CCN number concentration,

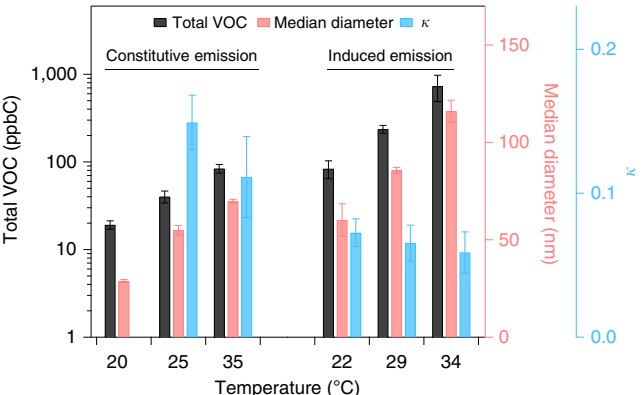

**Figure 3 | Effect of heat on emission amounts and particle properties.** The mixing ratios of total volatile organic compounds (VOC) from plant emissions (black bar, note log axis), median diameter of aerosol particles (red bar) and hygroscopicity parameter κ (blue bar) is plotted as a function of plant temperature. The constitutive emissions from unstressed pine trees and induced emissions from stressed mixed boreal trees are shown. The error bars represent the s.d. of the measurement (the detailed number of measurements included in Supplementary Table 4). For constitutive emissions at 20 °C, κ data are not available (noted by *) because particle sizes were too small (median diameter 26 nm) for cloud condensation nuclei (CCN) activation measurements.

demonstrating the important influence of the emission composition on the CCN activity of SOA.

The decrease of κ is related to the different chemical composition of the SOA from sesquiterpene-dominated emissions compared with that from monoterpene-dominated emissions. According to Petters and Kreidenweis[20] (equation (3), see Methods section), κ is inversely proportional to the molecular weight of the solute compound provided that all other parameters are constant. The average sesquiterpene oxidation products have higher molecular weight than those from monoterpene oxidation due to higher carbon number and higher molecular weight of sesquiterpenes (sesquiterpene 204 g mol$^{-1}$ versus monoterpene 136 g mol$^{-1}$)[21–23]. Therefore, κ is expected to be lower for SOA from emissions with higher sesquiterpene fractions. κ of SOA from monoterpene- and sesquiterpene-dominated emissions in our study are generally consistent with κ of SOA from the oxidation of single monoterpenes and single sesquiterpenes[21,24–28] (Supplementary Tables 1 and 2), considering the difference in VOC and experimental conditions which can cause variations of κ. We conclude that the presence of induced sesquiterpene emissions lowers κ.

**Effects of heat and drought.** In contrast to biotic factors, abiotic factors such as mild heat (up to 35 °C) did not significantly change the relative contributions of the different VOC classes (monoterpenes, sesquiterpenes and others, as described in Method) for both constitutive emissions (Supplementary Fig. 2) and induced emissions (Supplementary Fig. 3). Within each class, the contribution of some individual compounds changed, most distinct in the 'others' class, because specific compounds respond to temperature changes differently (cf. Supplementary Fig. 4). However, even mild heat increased the total VOC emissions for both types of emissions substantially (Fig. 3), consistent with previous studies[9,29]. Accordingly, heat led to larger SOA particles under similar photooxidation conditions, but the κ values of the SOA remained relatively invariant as temperature changed, with a seemingly small decreasing trend with increasing plant temperature. For the

SOA from constitutive emissions, the difference of $\kappa$ between 25 and 35 °C are not statistically significant (t-test, $P = 0.09$) and for the SOA from induced emissions, the difference in $\kappa$ is significant only for the entire temperature interval of 22–34 °C (t-test, $P = 0.017$, Fig. 3). The latter may be partly due to the detailed changes in the 'others' class or minor variations in the contributions of specific VOC classes. In this study, the particles were formed by homogeneous nucleation which means that particle number and particle size increased with increasing emissions. In the presence of pre-existing particles, only the size of the particles will be affected.

The CCN activity of particles is determined by their size and hygroscopicity[7,11,30]. The effect of elevated plant temperature on particle size was dramatic due to the substantial increase in the amount of VOC emitted and thus their oxidation products, which enhanced particle growth by condensation. However, the effect of elevated plant temperature on $\kappa$ itself was small, which is attributed to the overall little change in the emission composition. $\kappa$ determines the critical diameter for CCN activation. Larger aerosol size at constant $\kappa$ implies that more particles will be activated and more cloud droplets will be formed (see also Supplementary Fig. 5). Moreover, the effect of mild heat on particle size and $\kappa$ is independent whether the emissions are constitutive or induced. The effect of heat in the case of induced emissions by biotic stress represent the effect of co-occurring stressors, a common situation for plants under natural conditions[31].

In addition to heat, we also investigated the effect of water shortage (drought) on plant VOC emissions. Similar to the effect of heat, we found that with monoterpenes dominating the total emissions ($> 80\%$), the general emission composition of a pine did not change much with drought (Supplementary Fig. 6a). However, drought decreased the total amount of emissions (by up to $\sim 30\%$, Supplementary Fig. 6b). The magnitude of the response of VOC emissions to drought was relatively insensitive compared with heat. We conclude that decreasing VOC emissions by drought should affect the CCN activity of SOA in the opposite way as heat, and the overall effect would be a smaller activated fraction and less cloud droplets.

## Discussion

Biotic and abiotic stressors affect VOC emissions of plants and therefore the resulting SOA. Increased temperatures increase VOC emissions[9,29] promoting particle growth leading to a higher number of particles larger than 100 nm, that is, higher fractions of CCN active particles in the boundary layer[12,32–34]. Biotic stresses enhances the SOA mass concentration[16–18] and a model study assuming an increase of monoterpene emissions indicates higher aerosol and CCN concentrations in forests influenced by insect outbreaks[17]. The results, presented here for the first time, directly show how environmental conditions of plants can affect the CCN activity of biogenic SOA via the plants' emissions: biotic stresses by causing induced VOC emissions, modifying hygroscopicity, or abiotic stresses by changing the amount of constitutive VOC emissions, affecting particle size distribution.

Our findings provide important information on the potential impact of vegetation on the CCN activity of biogenic SOA, and thus on cloud properties. They are one puzzle piece for understanding the complex coupling between terrestrial plants and climate change. Climate change induces both long term, slow changes in the climate parameters such as global mean temperature change and short term episodic changes, such as heat and water shortage extremes. Plants can adapt to slow long-term climate changes to a varying extent via

phenological changes, evolutionary and genetic changes and migration[15]. Plant species with long lifetime such as trees have limited capability to adapt[15] and if climate zones move faster than vegetation zones, as currently observed[15], more trees will experience stresses more often. Furthermore, with climate change, the frequency of unfavourable environmental conditions such as heat, droughts, and infestation are projected to increase in many regions[3]. Our study mainly considered mild abiotic stresses, excluding extreme heat waves and severe droughts. In this case, heat enhances the CCN activity of particles by forming larger SOA particles, which also scatter more solar radiation[35]. Insect infestation of plants is expected to become more prevalent since herbivorous insects survive better in warming climate[15], adding to the fraction of forest being already affected by infestation at present[16]. These biotic stresses lead to induced VOC emissions which in case of sesquiterpenes decrease CCN activity. However, these changes in $\kappa$ can be compensated by higher SOA yields of induced VOC[18].

This study shows that environmental factors have an important impact on ambient CCN concentrations and properties of SOA. The effects of induced emissions should be considered in models that simulate the CCN concentrations and the impact on climate. Neglecting such effects can lead to significant biases. For example, models often simulate the CCN concentrations assuming a constant $\kappa$ value of $\sim 0.1$ for all organic aerosols regardless of the source. However, a study showed that a change of $\kappa$ of SOA by 50% (from 0.14 to 0.07 and 0.21) affects the CCN number concentration by $\sim 40\%$ (ref. 36). To demonstrate the potential effect of $\kappa$ change obtained here, we applied a decrease of $\kappa$ due to biotic stresses to a typical particle size distribution dominated by organic components as observed over a boreal forest in Finland as shown in Fig. 4. Our calculations show that if the plant emissions were dominated by sesquiterpenes instead of monoterpenes, the CCN number

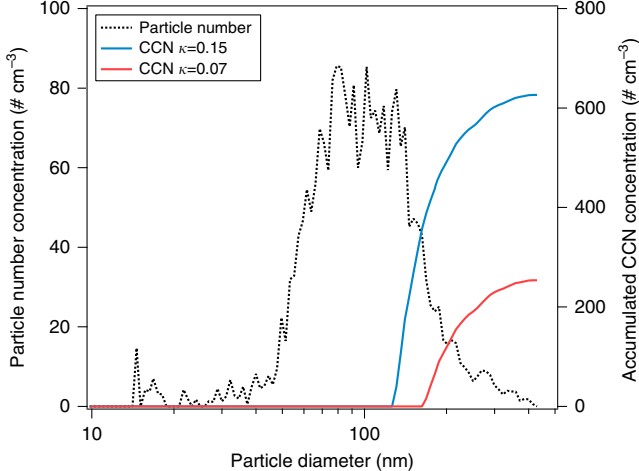

**Figure 4 | Impact of hygroscopicity parameter $\kappa$ changes on cloud condensation nuclei concentrations.** Measured ambient particle number size distribution (black dotted line, left axis) and the derived accumulated cloud condensation nuclei (CCN) number size distribution (right axis) are shown. The particle size distribution was measured in a boreal forest near Jämijärvi, Finland in May 2013 at a period when organics dominated the total aerosol mass ($> 80\%$). The CCN concentration was obtained considering 0.2% supersaturation, a typical supersaturation in clouds, using $\kappa$ of 0.15 (blue line) and 0.07 (red line) corresponding to the value for SOA from monoterpene-dominated and sesquiterpene-dominated emissions obtained in our plant chamber study shown above. Note the log $x$ axis.

concentration would reduce by ∼60% at 0.2% supersaturation. This suggests that biotic stresses can have significant influence on the CCN concentration in areas where biogenic SOA components dominate the particle composition. Currently, there are no direct field measurements reporting the effects of biotic stress on the CCN activity of biogenic SOA and CCN number concentrations, to our knowledge. Future field measurements of CCN in periods when biotic stresses induced emissions are dominant will help to assess the impact of biotic stresses on CCN activity and concentration.

The specific impacts on the ambient CCN number concentration and cloud formation depends on parameters such as ambient particle size distribution, aerosol composition, mixing state and supersaturation in clouds. Moreover, more biotic and abiotic stressors than discussed here may affect plants' emissions[8,37] and thus the composition and CCN activity of SOA. For example, mechanical stress due to a passing storm can induce higher monoterpene emissions from pine[38,39], which should affect CCN activity similar to heat but in a more quick and episodic way. We investigated young boreal tree species. It is possible that different plant species with different ages may exhibit different responses in VOC emissions to stresses and thus further change SOA composition and properties. For example, simulated herbivory on different tree species has been shown to cause different responses in VOC emissions, which alters the SOA composition with some variability[40,41]. Because various plant species with different ages as well as various environmental factors are involved and multiple factors can have synergistic or counteracting effects, the overall impacts are complex and cannot be assessed quantitatively here. Nevertheless, we provide a general scheme derived from our findings of the effects of biotic stresses, heat and drought on particle size, $\kappa$ and CCN number concentrations in Supplementary Fig. 7. Our results show the potential importance of induced emissions for SOA acting as CCN. Here for the constitutive emissions, heat and drought cause a 27% increase or a 37% decrease in the CCN number concentration, respectively. The biotic stress alone caused a 47% increase while biotic stress plus heat caused a 93% increase in the CCN number concentration compared with the reference case of constitutive emissions at room temperature (right lower white dot in Supplementary Fig. 7). Despite the complexity, the effects of various stresses, especially understudied biotic stresses, should be further investigated in laboratory and field studies as well as integrated into comprehensive models to better represent the feedbacks between terrestrial plants and climate.

## Methods

**Experimental setup and procedure.** The experiments were conducted in the Jülich Plant Atmosphere Chamber (JPAC). VOC emitted from boreal forest trees (pine (Pinus sylvestris L.), spruce (Picea abies L.), and birch (Betula pendula L.)) in a plant chamber were fed into a separate reaction chamber and were degraded by photooxidation to form SOA via homogeneous nucleation. JPAC was optimized to generate and investigate SOA by oxidation of VOC emissions from plants. The setup used here has been described in details by Mentel et al[19]. and was used previously to study SOA formation and properties[18,19,42–48]. A schematic representation of the setup is shown in Supplementary Fig. 1.

JPAC consists of three large glass chambers (0.164, 1.150, and 1.450 m$^3$), each in a separate temperature-controlled housing and operated as a continuously stirred tank reactor. The 1.450 m$^3$ chamber served as reaction chamber (RC), while the smaller chambers were used to host the plants, denoted as plant chambers (PC, PC1 for the 1.150 m$^3$ chamber and PC2 for 0.164 m$^3$ chamber). VOC emitted from trees in a plant chamber were fed into the reaction chamber and were degraded by photooxidation to form SOA via homogeneous nucleation. The PC were illuminated with discharge lamps (HQI 400 W/D; Osram, Munich, Germany), which simulate the solar spectrum reaching photosynthetic photon flux densities (PPFD) of up to 480 µmol m$^{-2}$ s$^{-1}$ (PC1) and 700 µmol m$^{-2}$ s$^{-1}$ (PC2) at full illumination. Switching on and off these visible-light lamps provided a day-night cycle for the plants. Purified air, which was free of particles, VOC, NO$_x$, and ozone with around 350 ppm CO$_2$ added, flowed through the PC and transferred the VOC emitted by the plants to the RC. Besides the flow from the PC, two additional air

streams supplied the RC with ozone (≈90 ppb) and water vapor. By controlling the humidity in this air stream, the relative humidity (RH) in the RC was held at constant 65% at constant RC temperature of 17 °C. The residence time of the chambers were ∼20 min in the PC1, 5–8 min in the PC2 and approximately 65 min in the RC. Inside the RC, a UV lamp (Philips, TUV 40W, $\lambda_{max} = 254$ nm) was switched on for certain periods to produce OH radicals from ozone photolysis and the subsequent reaction of O ($^1$D) radicals with water vapor. In the RC, OH concentrations depended on the introduced VOC amount since O$_3$, H$_2$O and UV intensity, and thus OH production, were kept constant. OH concentrations, derived from an OH tracer (deuterated cyclohexane-d$_{10}$), typically ranged $2 \times 10^7$–$6.5 \times 10^7$ molecules per cm$^3$. This range is about an order of magnitude higher than that observed in the atmosphere in boreal regions during summer[49].

The experiments were conducted by applying the following cycle: first the visible light in the PC was turned on to initiate the diurnal cycle as described below. We waited until the RC reached a steady-state regarding VOC concentrations. The steady state also included the ozonolysis reactions of VOC since ozone was constantly added into the RC. After reaching the steady state, the UV lamp in the RC was switched on to generate OH radicals and to induce photochemical particle formation. Formation of particles was only observed when the UV lamp was switched on and OH radicals were produced, as observed previously[19]. Experiment start is defined as the time when the UV lamp was switched on.

In this study, a mixed seedling stand of two pines (Pinus sylvestris L.), one spruce (Picea abies L.), and one birch (Betula pendula L.) (3–4 years old, about 1.1–1.3 m high) was housed in the PC1. These trees are mainly monoterpene emitters under unstressed conditions[19]. They were stored outdoor under natural conditions in the Forschungszentrum Jülich campus located near a forest. These trees were found to be infested by aphids, although we could not differentiate the exact species. They were only slightly infested and visual check only showed very slight defoliation and foliage discolouration without any significant damages. Such insect infestation is a part of the natural conditions that plants are facing. The aphid infestation was consistent with the high sesquiterpene emissions[8,9,37] (Supplementary Fig. 3). Similar sesquiterpene dominated emission composition was observed reproducibly for different types of boreal trees on different individuals (pine, spruce and birch) and in different years. Therefore, such emission composition represents one typical emission composition when trees are under insect infestation. Regular forest inspections reveal that insect infestation is common to about 10% of boreal trees[14,16,50].

To simulate the diurnal cycle for the trees, at 02:00 UTC the lamps were turned on sequentially creating an artificial dawn of 1 h. After 15 h of full illumination, the lamps were turned off in the reverse fashion. This led to a dark period of 7 h for the plants. The temperature in the PC was varied between 22 and 34 °C to investigate effects of temperature on VOC emission strength and composition, especially mild heat conditions[51]. After setting a new temperature in the PC, the plants had 8–18 h to adjust to the new conditions. In addition, two experiments were conducted in which the lamps in the PC remained off during the day and mainly VOC out of storage pools of the plants were emitted[9]. For these emissions, fractions of monoterpenes and sesquiterpenes were between those of the monoterpene-dominated and sesquiterpene-dominated cases (denotes as 'intermediate' case) because light dependent sesquiterpene emissions decreased more than monoterpene emissions in the dark. All other experimental conditions except otherwise mentioned were kept the same to make the experiments comparable.

Trees in absence of biotic stress were studied in another series of experiments. A stand of eight pine trees situated in the PC1, a single spruce and a single birch situated in the PC2 were investigated separately for VOC emissions (Supplementary Fig. 2) and particle formation[19]. By raising the plant temperature, VOC emissions were increased and the CCN activity of particles produced from these tree emissions were investigated. In this series of experiments, the CCN data from the experiments with spruce and birch emissions are limited because particle sizes were small and particle numbers were low for CCN activity studies due to low VOC concentrations. Therefore mainly CCN data from pine are discussed. Similar to the stressed trees case, the temperature in the PC was varied between 20 and 35 °C to investigate effects of mild heat.

In addition, to investigate the effects of drought on pine VOC emissions, a single pine tree (unstressed) was placed in the PC2 and watering was stopped and restarted in the same way as described by Wu et al[52]. Briefly, in these experiments, a three to four year old Scots pine was exposed to a diurnal rhythm of 11 h illumination (06:00–17:00) and 11 h darkness, and a simulation of twilight of 1 h each in the morning and evening, respectively. The pine was exposed to several drought/watering cycles. The emission rates at the stable period during the day were used. Heat and drought are used as two examples of the various abiotic factors affecting plant VOC emissions[37,39,53–57].

**Instrumentation.** The VOC were monitored with two Gas Chromatography-Mass Spectrometry (GC-MS, Agilent) systems and with a Proton Transfer Reaction—Mass Spectrometer (PTR-MS, Ionicon) with the details described in Mentel et al[19]. and Kleist et al[9]. Briefly, GC-MS systems measured VOC at the outlet of the PC. One GC-MS system was optimized to measure VOC from C5 to C20 including isoprene, monoterpenes and sesquiterpenes as well as compounds from lipoxygenase activity (LOX products) or phenolic compounds such as methyl salicylate[58]. The second GC-MS system was optimized to measure short chained

oxygenated VOC from methanol up to $C_{10}$ compounds[59]. Calibration of both systems was conducted as described by Heiden *et al*[60]. The PTR-MS was used to determine the concentrations of VOC and oxidation products at a time resolution of 10 min and was switched continuously between the outlet of the PC and the outlet of the RC.

The number concentration and size distributions of particles were measured with an Ultrafine Condensation Particle Counter (CPC, TSI, Model 3025A) and a Scanning Mobility Particle Sizer (SMPS, TSI, Model DMA 3071 and CPC 3022A).

**VOC emission composition and related SOA composition.** Induced VOC emissions are *de-novo* emissions, that is, they are directly coupled to photosynthetic activity of the plants. Although VOC induced by stress depend on tree species and specific stressors, there are mainly three groups of compounds based on their biosynthesis pathways: terpenoids (such as monoterpenes, sesquiterpenes), $C_6$ lipoxygenase (LOX) products (also known as green leaf volatiles) and aromatic products of the shikimate pathway (for example, methyl salicylate (MeSA))[2]. Constitutive terpenoid emissions are also of *de novo* type. In case plants have storage pools (for example, resin ducts), they are able to store some compounds like monoterpenes. Compounds from pools are emitted by physical diffusion governed by the plant temperature. Because monoterpenes and sesquiterpenes are dominant components in the VOC emissions of this study, which together typically account for more than 90% of the total VOC emissions, the VOC are classified as three classes: monoterpenes, sesquiterpenes and others (including LOX products, oxygenated VOC, aromatic compounds, isoprene and so on). We use the notation 'emission composition' to refer to the relative contributions of different emission classes (monoterpenes, sesquiterpenes and others) in total VOC emissions.

On the basis of the particle yield data of monoterpenes and sesquiterpenes from plant emissions in our previous studies[18,19] (17% for sesquiterpenes and 5% for monoterpenes), the relative fraction of SOA components from monoterpenes and sesquiterpenes oxidation could be roughly estimated from the VOC emission composition. For example, assuming the total SOA mass can be predicted by a linear combination of SOA yields from each precursor, the mass ratio of the SOA components from monoterpenes to the SOA components from sesquiterpenes in the 'Intermediate' case in Fig. 2 is around 0.8.

**Droplet activation measurement.** The number concentration of activated particles was measured with a Cloud Condensation Nuclei Counter (CCNC, Droplet Measurement Technologies, CCN-100) for supersaturations (SS = RH-100%) between 0.17 and 1.1% with the setup described previously[61]. In parallel, the particle size distribution (15.1–399.5 nm) and total number concentration were measured with a SMPS system (TSI Model 3071 and CPC 3022A) and a water-based CPC (TSI, Model 3785), respectively. Before entering the instruments, the poly-dispersed aerosol was dried with a silica gel diffusion drier to RH <5% and then neutralized with a Kr-85 neutralizer (TSI Model 3077).

The activated fraction was calculated as the ratio of the number concentration of activated particles to the total particle number concentration. The calculated activated fraction was compared with the cumulative size distribution starting from the maximum diameter. The size, at which the cumulative particle size distribution was equal to the activated fraction, was defined as dry critical activation diameter ($D_{crit}$). It was assumed that the aerosol particles were internally mixed. This approach is applicable to the period when the nucleation already stopped i.e., we measured the CCN activity at the period when the conditions in the RC reached the steady state. Five different SS were set in the CCNC: the first step for 20 min, the others for 10 min each. To ensure stable conditions in the CCNC, only data from the last 6 min of each SS step were used to determine $D_{crit}$, which overlapped with three SMPS scans each lasting 2 min. For each set of SS, the $D_{crit}$ values derived from three size scans were averaged. The contribution of multiple charged particles was corrected with the measured size distribution assuming a natural charge distribution. Ammonium sulfate aerosol was used to calibrate the SS of the CCNC based on data sets in the literature[62] (OS1 data set therein).

From the $D_{crit}$ and SS data, the hygroscopicity parameter $\kappa$ was determined using the method in Petters and Kreidenweis[20]. Petters and Kreidenweis[20] developed a theory to parameterize CCN activity data using $\kappa$ based on Köhler theory[63]. $\kappa$ is a measure of the hygroscopicity of particles, which is defined in the following equation:

$$\frac{1}{a_w} = 1 + \kappa \frac{V_s}{V_w} \tag{1}$$

$a_w$, $V_s$ and $V_w$ are the water activity, volume of solute and volume of water in an activated droplet, respectively.

The following equation can be derived by using equation (1) in the original $\kappa$-Köhler theory.

$$S = \left(1 + \kappa \cdot \frac{D_{dry}^3}{D_p^3 - D_{dry}^3}\right)^{-1} \cdot \exp\left(\frac{4M_W \sigma_{sol}}{RT\rho_W D_p}\right) \tag{2}$$

$S$: saturation ratio, $S = SS + 1$;
$D_p$: droplet diameter;
$D_{dry}$: dry particle diameter;
$M_w$: molecular weight of water;
$\sigma_{sol}$: surface tension of droplet solution;
$\rho_w$: density of water.
$R$: gas constant ($8.314 \, J \, mol^{-1} \, K^{-1}$)
$T$: temperature.
$\sigma_{sol}$ is assumed to be equal to that of water. Although organics can partition to the droplet surface, using the surface tension of water to calculate $\kappa$ is a reasonable assumption for droplet at activation[64,65].

Comparing the $\kappa$ definition in the $\kappa$-Köhler theory to other parameterizations of water activity (namely van't Hoff factor approach) yields[20],

$$k = i \frac{M_w/\rho_w}{M_s/\rho_s} \tag{3}$$

where $\rho_s$ and $\rho_w$ are the density of solute and water, and $M_s$ and $M_w$ are the molecular weight of solute and water, respectively. $i$ is the van't Hoff factor. It is the actual number of molecules or ions produced per solute molecule when a substance is dissolved[66–68]. Since most organics do not dissociate, $i$ is close to 1. The variability of density of most organics in SOA is small and can be assumed to be constant. equation (3) shows that $\kappa$ is inversely proportional to the molecular weight of solute assuming other factors are relatively constant.

$\kappa$ of SOA formed from the photooxidation of VOC emitted by unstressed trees and stressed trees were compared using $t$-test. $\kappa$ of SOA formed at different plant temperature were also compared.

**Ambient particle measurement.** The ambient particle size distribution and chemical composition were measured in a boreal forest near Jämijärvi, Finland in May and June of 2013 during the PEGASOS (Pan-European Gas-AeroSOl-climate interaction Study) Finland campaign using instruments on board a Zeppelin-NT airship. The particle number and size distribution were measured using a CPC (TSI, model 3786) and SMPS (TSI, DMA 3081 and CPC 3786). The particle chemical composition was measured using a High-Resolution Time-of-Flight Aerosol Mass Spectrometer (HR-ToF-AMS, Aerodyne Research Inc.[69]) adapted to airship measurement[70]. The measured particle size distribution was used to derive the CCN number concentrations. The activation diameter ($D_{crit}$) was obtained at 0.2% supersaturation. In the calculation of CCN number concentration, the activation fraction was set to 0, 0.5 and 1, respectively, if $D_{crit}$ was above, within and below a size bin of SMPS.

**Modelling of the effects of stresses on CCN concentrations.** The effects of various biotic stress, heat and drought on and $\kappa$ and particle size, thus on CCN number concentration were conceptually modelled based on the results in this study as shown in Supplementary Fig. 7. The CCN number concentration was derived at 0.2% supersaturation using a typical particle size distribution in the boreal forest in Finland (*cf.* Fig. 4).

While for the SOA from constitutive emissions, heat and drought have little effect on $\kappa$, heat increases particle size and drought decreases particle size due to their effects on VOC emission strengths. Similarly, for SOA from induced emissions, heat also increases particle size while having little effect on $\kappa$. The particle size changes due to heat were calculated using SOA mass changes corresponding to the exponential changes of VOC emissions with temperature as derived from Fig. 3. A temperature increase of 3.7 °C (a global mean temperature in 2100 projected by IPCC report 2013 (ref. 3)) was used here. The particle size change due to drought was calculated using the SOA mass change corresponding to the total VOC changes in Supplementary Fig. 6. A cubic relationship between the particle mass and diameter was assumed in the calculations of the effects of these environmental factors on particle size. The effect of particle size (shown as median diameter) on CCN number concentration was obtained by shifting the size distribution in constant logarithmical diameter steps.

Induced emissions decrease $\kappa$ while increasing particle size due to the higher SOA yield of sesquiterpenes compared with monoterpenes[18]. The particle size change due to biotic stress induced emissions was calculated using the SOA mass corresponding to the SOA yields of sesquiterpene-dominated emissions (17%)[18] and monoterpene-dominated emissions (5%) and assuming the same total VOC emissions (ppbC).

**Data availability.** The data supporting the findings of this study are available on reasonable request to the corresponding author.

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

## Acknowledgements

We thank the funding support from the EUROCHAMP2 and PEGASOS under EC 7th framework. We thank the support of PEGASOS Zeppelin 2013 campaign team.

## Author contributions

D.F.Z., A.B. and T.F.M. wrote the manuscript. T.F.M., A.K.S. and J.W. organized and designed the laboratory experiments. T.F.M. organized the PEGASOS Zeppelin Campaign. D.F.Z., A.B., R.T., E.K., C.W., J.W., T.F.M. conducted the laboratory data collection and analysis. D.F.Z, R.T., F.R. conducted ambient data collection and analysis. D.F.Z., A.B., A.K.-S., Y.R., J.W., T.F.M. edited the manuscript. All authors discussed the results and commented on the paper.

## Additional information

**Competing financial interests:** The authors declare no competing financial interests.

**Publisher's note**: 

