## [Peer Review File · Nature Communications]

Reviewers' comments:

Reviewer #1 (Remarks to the Author):

A. The manuscript addresses whether the environmental changes causes changes in the emission of biogenic volatile organic compounds (BVOCs) that lead to changes in cloud formation via secondary organic aerosol acting as cloud condensation nuclei. The authors used laboratory chamber experiments to simulate the response of BVOC emissions from a mixture of boreal forest trees for two types of changes: constitutive emissions due to temperature and water availability changes and stressor-induced emissions especially due to insect infestation. The BVOCs emitted were monitored using two complementary analytical instruments: proton-transfer reaction mass spectrometry and gas chromatography/mass spectrometry. SOA and CCN were measured by a scanning mobility particle sizer and a cloud condensation nuclei counter, respectively. The experiments show that changes in temperature and water availability cause differences (but not necessarily statistically significant) in the total VOCs emitted by boreal forest trees, which has a small effect on aerosol hygroscopicity. Insect infestation stresses boreal forest trees differently, the composition of BVOC emissions changes, and for this case, causes enhanced sesquiterpene emission which leads to lower aerosol hygroscopicity.

B. The research is novel because it connects an understudied, yet likely important, climate feedback cycle. This work shows the potential for the environmentally induced and constitutive changes in BVOC emissions to lead to changes in SOA and CCN hygroscopicity, and demonstrates the need to consider these factors in climate models. The latter point is especially well-explained on p. 9 (first full paragraph).

C. The experimental approach is appropriate for the work. One question that the authors should consider is whether the relatively young trees (3-4 years) used in this work would respond similarly to drought, temperature changes, and aphid infestation as mixed-age or mature population (as found in boreal forests) would respond. While it is beyond the scope of this study to study all age ranges and conditions, a comment to address this uncertainty would be helpful.

D. The use of statistics and treatment of uncertainties is appropriate. The manuscript would be improved if the number of replicates used to calculate the standard deviations for the error bars in Figures 2 and 3 (most easily added the captions) were included.

E. The conclusions are valid. The authors explained the limitations of the current work especially in terms of the how temperature and drought impacts this particular boreal forest trees, how the insect infestation used in this work (aphids) is a single example, and that different plants with different insect interactions may create different SOA, which in turn change the hygroscopicity of CCN. This is an important point and it could be strengthened with an example (perhaps review Faoula et al. *Atmos. Chem. Phys.*, 2015, which discussed SOA changes due to simulated herbivory in several plant species)

F. Beyond the suggestions included in this review, the abstract tends to overgeneralize the applicability of this work. It would helpful to acknowledge (in the abstract) the conditionality of this work, i.e. the scope of this work is one group of boreal forest trees and one type of insect infestation.

G. The references are appropriately applied and cited.

H. The text is mostly readable with the exception of a few sentences that are inconsistent with scientific style and the bulk of the text. For example, in paragraph 1, the use of the imperative in "Note that already nowadays environmental stress factors are affect plants" is awkward. In the following sentence "...more than 40% of forest trees in Europe suffer from various stresses in which biotic stresses account for ~40%" of what? The total stresses? Rewording this sentence would improve clarity. Also "nowadays" is a casual and vague term (used also p. 9).

On page 8 the sentence "Climate changes induces both long terms changes and short term changes in the climate parameters" needs to be reworded to improve clarity.

Also on page 8 the following sentence "...more trees will be exposed more often to stresses." may be improved by rewording "...more trees will experience stresses more often."

Reviewer #2 (Remarks to the Author):

Review on "Environmental conditions regulate the impact of plants on cloud formation" by Zhao et al.

This is an interesting paper on a timely, appealing and important topic. However, I think it is in the limit of acceptance for a high profile journal of the Nature family because it does not seem robust, comprehensive, quantitative and novel enough.

See my comments while reading the paper:

Abstract too verbose and with some confusing sentences like for example: "...biotic and abiotic environmental factors that regulate emissions of VOC also modify the CCN activity of the resulting SOA". Does this occur because of a direct effect of those environmental factors or only through the changes they generate on SOA?

Or the sentence starting with "Our findings" better state which are they? It is not clear.

Or in introduction: For example it is not only "nowadays" that environmental stress factors are affecting plants....this has occurred always.

Or the sentence "...This implies a strong feedback between plants' emissions and climate ". This is still not demonstrated at least its actual quantitative importance.

Or the sentence " effects on VOCs emission of plants are much less investigated than those on net CO₂ exchange..." It is true but avoid giving the wrong impression that there is not much on this, since in fact, there are multitude of studies in this area.

The statistics are not clear nor robust either. For example, there is no variance data presented for monoterpene/sesquiterpene ratio or no "n" in the figure 2 caption

The experimental design: tested hygroscopicity of SOA are in a quite narrow range. One wonders what would happen in a wider range.

Different compounds have different responses to temperature, so I wonder how this was not appreciated with increasing temperatures tested. How there was no change in emission pattern for constitutive and even more for induced emissions? Any explanation for that?

Figure 3. The caption statement on significance of changes in k does not seem to be appreciated in the figure. Unclear.

Water shortage (drought) appear in page 8 as a surprise not presented, nor treated, before. By the way, this paragraph on drought needs more work and detail.

In the following paragraph, more than modeled values the reader is interested in actual measurements ...not possible?

Next paragraph: I agree this is an important study with important and interesting information such the one on biotic stresses likely having significant influence on the CCN concentration in areas where biogenic SOA components dominate particle formation..... but this, by itself, is not completely new (this same research group itself has studied this for long), not very robust, not very extensive, not wholly convincing, not with real measurements, and specially with not much quantitative information, at least for a paper in a high profile journal like this one. It seems instead very adequate for a good atmospheric journal.

Figure 4 caption....why modeled and not measured values?

Experimental design-Extended figure 1. Why not using a single chamber where plants are in contact with ozone as it in fact occurs in nature?

Try to explain more clearly the caption of figure extended data figure 2.

Extended data figure 4: I am still surprised of this absence of change with temperature, and still would very much recommend having the same graph for the different monoterpene compounds and the different sesquiterpene and others compounds since they have different physic-chemical properties and temperature sensitivities

Extended data figure 6. As in most parts of the text I miss statistics here. How significant are the changes? They do not seem very strong or significant.

*Reviewers' comments:*

*Reviewer #1 (Remarks to the Author):*

*A. The manuscript addresses whether the environmental changes causes changes in the*
*emission of biogenic volatile organic compounds (BVOCs) that lead to changes in cloud*
*formation via secondary organic aerosol acting as cloud condensation nuclei. The authors*
*used laboratory chamber experiments to simulate the response of BVOC emissions from a*
*mixture of boreal forest trees for two types of changes: constitutive emissions due to*
*temperature and water availability changes and stressor-induced emissions especially due to*
*insect infestation. The BVOCs emitted were monitored using two complementary analytical*
*instruments: proton-transfer reaction mass spectrometry and gas chromatography/mass*
*spectrometry. SOA and CCN were measured by a scanning mobility particle sizer and a*
*cloud condensation nuclei counter, respectively. The experiments show that changes in*
*temperature and water availability cause differences (but not necessarily statistically*
*significant) in the total VOCs emitted by boreal forest trees, which has a small effect on*
*aerosol hygroscopicity. Insect infestation stresses boreal forest trees differently, the*
*composition of BVOC emissions changes, and for this case, causes enhanced sesquiterpene*
*emission which leads to lower aerosol hygroscopicity.*

*B. The research is novel because it connects an understudied, yet likely important, climate*
*feedback cycle. This work shows the potential for the environmentally induced and*
*constitutive changes in BVOC emissions to lead to changes in SOA and CCN hygroscopicity,*
*and demonstrates the need to consider these factors in climate models. The latter point is*
*especially well-explained on p. 9 (first full paragraph).*

*C. The experimental approach is appropriate for the work. One question that the authors*
*should consider is whether the relatively young trees (3-4 years) used in this work would*
*respond similarly to drought, temperature changes, and aphid infestation as mixed-age or*
*mature population (as found in boreal forests) would respond. While it is beyond the scope of*

*this study to study all age ranges and conditions, a comment to address this uncertainty*
*would be helpful.*

**Response:**

We thank the reviewer for the supporting remarks.

We have accepted the suggestion and made a remark on the use of young rather than more
mature trees. Indeed, the effect of age on how trees respond to stresses is understudied and
largely unknown. We agree with the reviewer that it is outside the scope of this paper. We
therefore added the following remark to better indicate the limitations of our statements in the
revised manuscript (now page 9):

“We investigated young boreal tree species. It is possible that different plant species with
different ages may exhibit different responses in VOC emissions to stresses and thus further
change SOA composition and properties.”

*D. The use of statistics and treatment of uncertainties is appropriate. The manuscript would*
*be improved if the number of replicates used to calculate the standard deviations for the*
*error bars in Figures 2 and 3 (most easily added the captions) were included.*

**Response:**

We accepted this comment. In the revised manuscript, we have added the number of
measurements used to calculate the standard deviations in the captions of Figure 2 and
separately in two newly added tables for Figure 2 and Figure 3 (now Extended Data Table. 1
and Table 2).

*E. The conclusions are valid. The authors explained the limitations of the current work*
*especially in terms of the how temperature and drought impacts this particular boreal forest*
*trees, how the insect infestation used in this work (aphids) is a single example, and that*
*different plants with different insect interactions may create different SOA, which in turn*
*change the hygroscopicity of CCN. This is an important point and it could be strengthened*
*with an example (perhaps review Faoula et al. Atmos. Chem. Phys., 2015, which discussed*
*SOA changes due to simulated herbivory in several plant species).*

**Response:**

We thank the reviewer for the supporting remarks and have accepted the suggestion. In the
revised manuscript, we have added an example to strengthen our discussion as follows.

“We investigated young boreal tree species. It is possible that different plant species with
different ages may exhibit different responses in VOC emissions to stresses and thus further
change SOA composition and properties. For example, simulated herbivory on different tree
species has been shown to cause different responses in VOC emissions, which alters the SOA
composition with some variability^{33,34}.”

*F. Beyond the suggestions included in this review, the abstract tends to overgeneralize the*
*applicability of this work. It would helpful to acknowledge (in the abstract) the conditionality*
*of this work, i.e. the scope of this work is one group of boreal forest trees and one type of*
*insect infestation.*

**Response:**

In the revised manuscript we have modified the abstract to fix the overgeneralization. We
clearly state that the study was based on one group of boreal forest trees and one type of
insect infestation (aphid here).

*G. The references are appropriately applied and cited.*

*H. The text is mostly readable with the exception of a few sentences that are inconsistent*
*with scientific style and the bulk of the text. For example, in paragraph 1, the use of the*
*imperative in "Note that already nowadays environmental stress factors are affect plants" is*
*awkward. In the following sentence "...more than 40% of forest trees in Europe suffer from*
*various stresses in which biotic stresses account for ~40%" of what? The total stresses?*
*Rewording this sentence would improve clarity. Also "nowadays" is a casual and vague term*
*(used also p. 9).*

*On page 8 the sentence "Climate changes induces both long terms changes and short term*
*changes in the climate parameters" needs to be reworded to improve clarity.*

*Also on page 8 the following sentence "...more trees will be exposed more often to stresses."*
*may be improved by rewording "...more trees will experience stresses more often."*

**Response:**

We thank the reviewer for the helpful comments pointing out our unclear sentences.

In the revised manuscript, we have accepted the suggestions and modified these sentences to
make them clear.

The sentence "Note that already nowadays environmental stress factors are affect plants" now
reads:

“Already at present, environmental stress factors strongly affect plants^{14, 15}.”

The sentence "...more than 40% of forest trees in Europe suffer from various stresses, in
which biotic stresses account for ~40%." now reads:

“...more than 40% of forest trees in Europe suffer from various stresses where biotic stresses
account for ~40% of the total stresses^{14,16}.”

The word “nowadays” in page 9 has been changed to “at present”.

The sentence "Climate changes induces both long terms changes and short term changes in
the climate parameters" have been reworded to improve its clarity. Now it reads:

“Climate change induces both long term, slow changes in the climate parameters such as
global mean temperature change and short term episodic changes, such as heat and water
shortage extremes.”

The sentence "...more trees will be exposed more often to stresses" on page 8 has been
modified to "...more trees will experience stresses more often" in the revised manuscript.

*Reviewer #2 (Remarks to the Author):*

*Review on "Environmental conditions regulate the impact of plants on cloud formation" by*
*Zhao et al.*

*This is an interesting paper on a timely, appealing and important topic. However, I think it is*
*in the limit of acceptance for a high profile journal of the Nature family because it does not*
*seem robust, comprehensive, quantitative and novel enough.*

**Response:**

We thank the reviewer for carefully reviewing our manuscript and giving constructive
comments. Based on these comments, we have substantially modified our manuscript. All the
comments have been addressed and we believe that these revisions have substantially
improved the manuscript. In the following, we provide the one-by-one responses to the
comments and the corresponding changes to the manuscript. The original comments are
shown in italics.

*See my comments while reading the paper:*

*Abstract too verbose and with some confusing sentences like for example: "...biotic and*
*abiotic environmental factors that regulate emissions of VOC also modify the CCN activity of*

*the resulting SOA". Does this occur because of a direct effect of those environmental factors*
*or only through the changes they generate on SOA?*

**Response:**

In the revised manuscript, we have omitted this sentence by condensing the entire abstract.

*Or the sentence starting with "Our findings" better state which are they? It is not clear.*

**Response:**

In the revised manuscript, we have modified this sentence to make it clear. Now it reads:

"The coupling of plant stresses, VOC composition and CCN activity points to an important
impact of induced plant emissions on cloud formation and climate."

*Or in introduction: For example it is not only "nowadays" that environmental stress factors*
*are affecting plants....this has occurred always.*

**Response:**

In the revised manuscript, we have modified this sentence. Now it reads:

"Already at present, environmental stress factors strongly affect plants^{14,15}."

*Or the sentence "...This implies a strong feedback between plants' emissions and climate".*
*This is still not demonstrated at least its actual quantitative importance.*

**Response:**

In the revised manuscript, we have changed this sentence to make is more precise. Now it
reads:

"This implies a potential important feedback between plants' emissions and climate."

*Or the sentence "effects on VOCs emission of plants are much less investigated than those on*
*net CO2 exchange..." It is true but avoid giving the wrong impression that there is not much*
*on this, since in fact, there are multitude of studies in this area.*

**Response:**

We agree. In the revised manuscript, we have modified this sentence to avoid potential
misunderstanding. Now it reads:

"The effects of environmental factors on VOC emissions of plants have been investigated by
a number of studies, but less intensive than their effects on the net CO₂ exchange of

plants^{8,9,17}. Only few studies address effects of environmental factors on induced VOC
emissions and SOA formation^{13, 19}.”

*The statistics are not clear nor robust either. For example, there is no variance data*
*presented for monoterpene/sesquiterpene ratio or no "n" in the figure 2 caption.*

**Response:**

In the revised manuscript, we improved the statistical analysis. We have added error bars of
the ratio of monoterpene to sesquiterpene. We also now specify the number of measurements
used to calculate standard deviations in the captions of Figure 2 and in two newly added
tables for Figure 2 and Figure 3 (now Extended Data Table. 1 and Table. 2).

*The experimental design: tested hygroscopicity of SOA are in a quite narrow range. One*
*wonders what would happen in a wider range.*

**Response:**

The hygroscopicity parameter (κ) of SOA from different biogenic organic precursors and
under different reaction conditions, converges to a narrow range, and most studies reported κ
in the range of approximately 0.05-0.2, roughly varying around 0.1 (see references (Prenni et
al., 2007; Frosch et al., 2011; Lambe et al., 2011) and the references in Extended Data Table
4). We guess that this is what the reviewer meant by “quite narrow range”. The κ values of
SOA found in our study are consistent with this range. In our experimental design, we
systematically changed the environmental factors (biotically stressed versus non-stressed;
different temperatures and water content), and we did not intentionally narrow or widen the
range of hygroscopicity of SOA.

*Different compounds have different responses to temperature, so I wonder how this was not*
*appreciated with increasing temperatures tested. How there was no change in emission*
*pattern for constitutive and even more for induced emissions? Any explanation for that?*

**Response:**

The “emission pattern” in this study refers to “the relative contributions of different classes of
VOC” as stated in our manuscript. Different individual compounds are lumped into three
classes: monoterpenes, sesquiterpenes and others. We agree that the emissions of different
compounds may respond differently to changes in temperature. As a matter of fact, we
observed that within each class (e.g. sesquiterpenes or monoterpenes), the relative
contributions of some individual compounds can change with temperature (see newly added

Extended Data Fig. 5). However, the relative ratios of other compounds did not change much
with temperature (see newly added Extended Data Fig. 5). Overall the relative contributions
of total monoterpenes, sesquiterpenes and others remained largely stable as shown in
Extended Data Fig. 6. An explanation for the stability of the relative contributions may be
that all three classes of volatiles contain stress-induced emissions. For stress induced
emissions, temperature dependence can be very different from those for constitutive
emissions, because it is not only the temperature dependence of the enzymatic system
synthesizing the VOC that determines the temperature dependence but also the temperature
dependence of the biosynthetic pathway eliciting the stress induced emissions. Hence we
cannot provide a general explanation for our observation that the contributions of
monoterpenes, sesquiterpenes and other emissions did not change substantially with
temperature.

Our concept is to use plants as holistic VOC sources to overcome biases by synthetic VOC
mixtures in many previous laboratory studies. We do not intend to present a biological
explanation for such changes. The focus of our investigations is how real and exemplary
constitutive and induced VOC emissions and their changes due to stresses affect SOA
properties such as κ and finally affect cloud formation. κ changes clearly with the change of
the MT/SQT ratio and this result holds even when we do not have a clear answer to the
question why the MT/SQT ratio was not substantially changed by temperature but clearly
changed by insect infestation.

In the revised manuscript, we have added the following explanation to clarify this point and
added an additional figure in the supplement (newly added Extended Data Fig. 5). In
addition, in order to avoid potential misunderstanding, we have changed the wording “VOC
emission pattern” to “VOC emission composition” throughout our manuscript.

“In contrast to biotic factors, abiotic factors such as mild heat (up to 35 °C) did *not*
significantly change the relative contributions of different VOC classes (monoterpenes,
sesquiterpenes and others, as described in Method) for both constitutive emissions (Extended
Data Fig. 3) and induced emissions (Extended Data Fig. 4). *Within* each class, the
contribution of some individual compounds changed, most distinct in the “others” class,
because specific compounds respond to temperature changes differently (c.f. Extended Data
Fig. 5).”

*Figure 3. The caption statement on significance of changes in k does not seem to be*
*appreciated in the figure. Unclear.*

**Response:**

In the revised manuscript, we have added p values and number of measurements for
calculating standard deviations (newly added Extended Data Table. 2) for the statistics result
in order to make it clear. Now it reads:

“The change of κ is not statistically significant in the constitutive emission case (t-test,
$p=0.09$, for the case of 25 °C and 35°C) and is only significant from 22 °C to 34 °C in the
induced emission case (t-test, $p=0.017$). The error bars represent the standard deviations of
the measurement (with symmetric positive and negative values, and the detailed number of
measurements included in Extended Data Table.2).”

*Water shortage (drought) appear in page 8 as a surprise not presented, nor treated, before.*
*By the way, this paragraph on drought needs more work and detail.*

**Response:**

In the revised manuscript, we have introduced the effect of drought early in the manuscript
(page 2, lines 60-62).

“We investigated the effect of aphid infestation as an example of biotic stresses and the effect
of heat and drought as examples of abiotic stresses for both constitutive emissions and
induced emissions.”

And in the revised manuscript, we have modified this paragraph further by adding more
details including one additional figure panel (now Extended Data Fig. 6a). Now it reads:

“In addition to heat, we also investigated the effect of water shortage (drought) on
plant VOC emissions. Similar to the effect of heat, we found that with monoterpenes
dominating the total emissions (>80%), the general emission composition of a pine did not
change much with drought (see Extended Data Fig. 6a). However, drought decreased the total
amount of emissions (Extended Data Fig. 6b). The magnitude of the response of VOC
emissions to drought was relatively insensitive compared to heat. We conclude that
decreasing VOC emissions by drought should affect the CCN activity of SOA in the opposite
way as heat, and the overall effect would be a smaller activated fraction and less cloud
droplets.”

*In the following paragraph, more than modeled values the reader is interested in actual*
*measurements ...not possible?*

**Response:**

We guess that the reviewer referred to the paragraph discussing Fig. 4. The aim here is to
assess how much the changes of hygroscopicity (κ) of biogenic SOA due to biotic stresses
found in our laboratory study can impact the CCN number concentration in the ambient -
from the microphysical point of view. Since besides hygroscopicity, also particle size affects
the CCN number concentration, the size distribution needs to be kept constant in order to
demonstrate the impact of changes of hygroscopicity (κ). Therefore, we derived the CCN
number concentration from a given constant size distribution using different κ values
representing different scenarios.

In field measurements, it is difficult to apportion the changes of particle size to specific
sources, e.g. induced emissions. One reason is that pre-existing aerosols in the field
measurement which also contribute to CCN number concentration can also vary, making it
difficult to specifically extract the effect of various stresses.

Admittedly, both the field measurement and model studies on CCN activity and number
concentration are important to understand the impact of stresses of plant on cloud formation
ability of biogenic SOA.

In the revised manuscript, we have added a brief discussion on this.

“Currently, there are no direct field measurements reporting the effects of biotic stress on the
CCN activity of biogenic SOA and CCN number concentrations, to our knowledge. Future
field measurements of CCN in periods when biotic stresses induced emissions are dominant
will help to assess the impact of biotic stresses on CCN activity and concentration.”

*Next paragraph: I agree this is an important study with important and interesting*
*information such the one on biotic stresses likely having significant influence on the CCN*
*concentration in areas where biogenic SOA components dominate particle formation..... but*
*this, by itself, is not completely new (this same research group itself has studied this for*
*long), not very robust, not very extensive, not wholly convincing, not with real measurements,*
*and specially with not much quantitative information, at least for a paper in a high profile*
*journal like this one. It seems instead very adequate for a good atmospheric journal.*

**Response:**

We would like to emphasize the novelty of this study with the background of previous
studies. Although there are multitude of studies on the effect of various stresses on VOC
emissions, there are only very few studies investigating the particle formation and SOA
properties formed from real plant VOC emissions including the studies of our group.
Importantly, no studies have directly investigated the CCN activation, the important
microphysical properties for climate, as a function of various stresses, to the best of our
knowledge.

Moreover, we guess the reviewer referred to field studies with the statement “not with real
measurements”. Our study is a laboratory study, which investigates the quantitative impacts
of various stresses under well-controlled conditions using real plants as representative,
holistic VOC sources. Our concept of laboratory studies is complementary to field studies as
it allows repeatedly investigating VOC mixtures from complex sources under well-controlled
physical conditions. (It is also complementary to classical laboratory studies as we use
complex real VOC sources rather than using single VOC or simple, artificial mixtures). We
studied a key quantity which is one prerequisite for vegetation-climate interactions: the cloud
droplet activation of the resulting SOA. In this sense we understand our manuscript - in its
inherent limitations- as also a trigger to promote more studies in the area including field
studies - a scope of Nature Communications. Our studies are necessary and helpful to direct
field studies, where many environmental factors (temperature, solar radiation intensity,
ambient oxidants concentrations, background aerosol type and concentrations etc.) affect the
VOC emission and CCN activation and the system is even more underdetermined.
Underdetermination and limited reproducibility make it extremely complicated to extract
quantitative effects of certain stresses on CCN activity and number concentrations. In
contrast, in well-controlled laboratory studies, we investigated the effects of certain stresses
systematically and quantitatively without interferences from other factors. We agree with the
reviewer in that we cannot and therefore did not, directly extrapolate our results to general
ambient cases. We did point out the principle importance based on the measurements, though.
This study clearly shows that environmental factors that affect plant emissions eventually
propagate all the way to CCN activity of SOA formed from these plant emissions and once
more highlights the importance of environmental conditions in the potential impact of
terrestrial vegetation on cloud formation and climate. An important conclusion from this
study is hence, that such effects should be included in climate models and are worth further

studies. We therefore debate that this makes our manuscript unsuited for publication in
Nature communications, especially as we now address the well taken major critics.

In addition, regarding “quantitative information”, we directly quantified for the first time
those parameters: changes of hygroscopicity (κ) of SOA due to insect infestation and heat.
These are the complete data set achievable in laboratories studies. And we further quantified
the effects of the biotic stresses on the CCN number concentrations in boreal forest areas
dominated by biogenic SOA. Admittedly, due to the complexity of the various environmental
stresses, “the overall impacts are complex and cannot be assessed quantitatively here” (as we
stated) or by any single study.

In the revised version, we clarified the manuscript, made the statistics more robust and
described our findings more quantitatively (see the changes based on each comments). We
wish that - given the clarifications and additional data - now the reviewer can kindly accept a
certain inherent lack of immediate generalization. This indeed will need field measurements,
which are beyond the scope of this study. Still our study represents an important step to better
understand these impacts.

*Figure 4 caption....why modeled and not measured values?*

**Response:**

Please see our responses to the similar question above (page 9, lines 243-244). The particle
size distribution was taken from our observations within the PEGASOS campaign near
Hyytiälä, Finland, in order to demonstrate the effect of our findings regarding size and κ to a
realistic situation. During this field campaign no CCN measurements were done.

*Experimental design-Extended figure 1. Why not using a single chamber where plants are in*
*contact with ozone as it in fact occurs in nature?*

**Response:**

There are three main reasons to use one plant chamber and one reaction chamber.

First, we need to determine VOC emissions from plants and initial concentrations of VOC for
SOA formation in the photochemical reaction. The initial VOC concentrations are essential to
determine the SOA yield (SOA mass formed per mass of reacted VOC). If there were only a
single chamber with plants and oxidants such as O_3 and OH together, neither real VOC
emissions nor the initial VOC concentrations in the reaction can be determined due to the
reaction loss with OH and O_3 .

Second, the UV light used in the laboratory setup to simulate photochemical reactions and
SOA formation can damage plants, which can confound the effects of the studied conditions
on plants.

Third, the plant chamber itself cannot provide the enough residence (reaction) time that is
needed for SOA formation and particle growth from small to large. In order to avoid water
vapor condensing caused by transpiration in the plant chamber, the flow rate needs to be
high. This reduces the residence time since residence time is inversely proportional to the
flow rates.

*Try to explain more clearly the caption of figure extended data figure 2.*

**Response:**

We have improved the caption to explain the caption more clearly. We have also added
another panel to explain the critical activation diameter and split previous panel b into two
panels to make it more clear.

*Extended data figure 4: I am still surprised of this absence of change with temperature, and
still would very much recommend having the same graph for the different monoterpene
compounds and the different sesquiterpene and others compounds since they have different
physic-chemical properties and temperature sensitivities.*

**Response:**

Please see our response to the similar question above (page 6, lines 172-174).

In the revised manuscript, we have added the explanation of this point and have accepted the
reviewer's suggestion. We have added one figure in the Methods part to show relative
fractions of individual compounds of monoterpenes, sesquiterpenes and others. While relative
ratios of certain compounds can change with temperature, those for other compounds remain
largely invariant with temperature. The overall VOC emission composition regarding the
relative contributions of monoterpenes, sesquiterpenes and others did not change much with
temperature.

*Extended data figure 6. As in most parts of the text I miss statistics here. How significant are
the changes? They do not seem very strong or significant.*

**Response:**

The detailed effects of biotic stress, heat and drought shown in this figure was calculated
using the results found in our laboratory study. Therefore, there are no error bars on it. We
apologize for not clearly explaining this. In the revised manuscript, we have clearly described
it.

This figure mainly works as one example of the typical effects of various stresses on CCN
number concentration and the exact values are not our focus since they may depend on many
parameters and hypothesis used. But the changes on CCN number concentration (indicated
by the color bar) shown here are strong. For the constitutive emissions, the heat or drought
cause a 27% increase or a 37% decrease in the CCN number concentration, respectively. The
biotic stress alone causes a 47% increase while biotic stress plus heat causes a 93% increase
compared to the reference case of constitutive emissions at room temperature (right lower
circle).

In the revised manuscript, we have also added the numbers of changes in the caption.

**References**

Frosch, M., Bilde, M., DeCarlo, P. F., Juranyi, Z., Tritscher, T., Dommen, J., Donahue, N.
382 M., Gysel, M., Weingartner, E., and Baltensperger, U.: Relating cloud condensation nuclei
activity and oxidation level of alpha-pinene secondary organic aerosols, *J. Geophys. Res.-*
*Atmos.*, 116, D22212, 10.1029/2011jd016401, 2011.

Lambe, A. T., Onasch, T. B., Massoli, P., Croasdale, D. R., Wright, J. P., Ahern, A. T.,
Williams, L. R., Worsnop, D. R., Brune, W. H., and Davidovits, P.: Laboratory studies of the
chemical composition and cloud condensation nuclei (CCN) activity of secondary organic
aerosol (SOA) and oxidized primary organic aerosol (OPOA), *Atmos. Chem. Phys.*, 11,
8913-8928, 10.5194/acp-11-8913-2011, 2011.

Prenni, A. J., Petters, M. D., Kreidenweis, S. M., DeMott, P. J., and Ziemann, P. J.: Cloud
droplet activation of secondary organic aerosol, *J. Geophys. Res.-Atmos.*, 112, D10223,
10.1029/2006jd007963, 2007.

REVIEWERS' COMMENTS:

Reviewer #1 (Remarks to the Author):

The revisions to the manuscript were appropriate for the comments and suggestions that I (reviewer 1) made for the initial review.

Reviewer #2 (Remarks to the Author):

Although I am still surprised by the absence of changes in BVOC composition with warming and drought, I think the authors have satisfactorily answered our questions and solved our concerns. I think, as already said for the first version, this is an interesting paper showing how constitutive and induced BVOC emissions, and their changes due to stresses, affect SOA properties such as κ , and how they finally affect cloud formation. The authors have also interestingly concluded that decreasing VOC emissions by drought should affect the CCN activity of SOA in the opposite way as heat, resulting in less cloud droplets. Although the authors cannot directly extrapolate their results to general ambient cases, I agree their results are helpful to direct future field studies. They moreover highlight the importance of environmental conditions in the potential impact of terrestrial vegetation on cloud formation and climate and therefore the interest and need of such effects being included in climate models. I am happy to now recommend acceptance.

REVIEWERS' COMMENTS:

Reviewer #1 (Remarks to the Author):

The revisions to the manuscript were appropriate for the comments and suggestions that I (reviewer 1) made for the initial review.

Reviewer #2 (Remarks to the Author):

Although I am still surprised by the absence of changes in BVOC composition with warming and drought, I think the authors have satisfactorily answered our questions and solved our concerns. I think, as already said for the first version, this is an interesting paper showing how constitutive and induced BVOC emissions, and their changes due to stresses, affect SOA properties such as κ , and how they finally affect cloud formation. The authors have also interestingly concluded that decreasing VOC emissions by drought should affect the CCN activity of SOA in the opposite way as heat, resulting in less cloud droplets. Although the authors cannot directly extrapolate their results to general ambient cases, I agree their results are helpful to direct future field studies. They moreover highlight the importance of environmental conditions in the potential impact of terrestrial vegetation on cloud formation and climate and therefore the interest and need of such effects being included in climate models. I am happy to now recommend acceptance.

Response:

We thank the two reviewers for carefully reviewing our manuscript again and giving the supportive remarks.